# Intelligent Fault Diagnosis Method for Gearboxes Based on Deep Transfer Learning

**Zhenghao Wu, Huajun Bai, Hao Yan, Xianbiao Zhan, Chiming Guo and Xisheng Jia ***

Shijiazhuang Campus, Army Engineering University of PLA, Shijiazhuang 050003, China
* Correspondence: xs_jia2022@163.com

**Abstract:** The complex operating environment of gearboxes and the easy interference of early fault feature information make fault identification difficult. This paper proposes a fault diagnosis method based on a combination of whale optimization algorithm (WOA), variational mode decomposition (VMD), and deep transfer learning. First, the VMD is optimized by using the WOA, and the minimum sample entropy is used as the fitness function to solve for the K value and penalty parameter $\alpha$ corresponding to the optimal decomposition of the VMD, and the correlation coefficient is used to reconstruct the signal. Second, the reconstructed signal after reducing noise is used to generate a two-dimensional image using the continuous wavelet transform method as the transfer learning target domain data. Finally, the AlexNet model is used as the transfer object, which is pretrained and fine-tuned with model parameters to make it suitable for early crack fault diagnosis in gearboxes. The experimental results show that the method proposed in this paper can effectively reduce the noise of gearbox vibration signals under a complex working environment, and the fault diagnosis method of using transfer learning is effective and achieves high accuracy of fault diagnosis.

**Keywords:** whale optimization algorithm; variational mode decomposition; deep transfer learning; gearbox; fault diagnosis

## 1. Introduction

Gearbox is a key component of the transmission systems of large instruments, such as helicopters, cars, and fans. These pieces of equipment are constantly subjected to external weather and rain, as well as subjected to high-intensity loads for extended periods of time, resulting in frequent gearbox failures that disrupt normal operation and even cause economic losses and casualties. Being able to detect failures at an early stage can avoid catastrophic consequences. Therefore, intelligent fault diagnosis of gearboxes has significant research value [1–6].

Currently, vibration signals are commonly used for fault diagnosis in gearboxes. The actual operating environment of the gearbox is extremely severe, with constantly changing load conditions, resulting in an irregular vibration signal in the gearbox. Interference between the internal components of the gearbox causes the vibration signal to be nonlinear; therefore, the collected vibration signal contains a variety of complex noise components. Consequently, the use of vibration signals for noise reduction processing and fault diagnosis is a hot topic in contemporary research, and fruitful results have been obtained [7–9]. After the discovery of the empirical mode decomposition (EMD) method of noise reduction for nonstationary, nonlinear signals, EMD-like methods have been widely applied to signal noise reduction. For example, Abdelkader et al. [10] used the average energy to optimize the threshold operation for the intrinsic mode function (IMF) component of the EMD to achieve noise reduction, and the experiments verified that this noise reduction method is more effective and sensitive for the detection and diagnosis of rolling bearing faults. Liu et al. [11] used kurtosis to select the intrinsic mode function (IMF) component of the EMD as the main IMF function, and then filtered the main IMF function with an

impact dictionary, which can separate the high-frequency resonant component from the meshing harmonics and partial noise to achieve noise reduction. Gao et al. [12] used integrated evaluation and wavelet thresholding to select and process the IMF components decomposed by ensemble empirical mode decomposition (EEMD), and used simulation methods to verify the feasibility of the method used to extract valid information from the signal under high noise. Liu et al. [13] used the complementary ensemble empirical mode decomposition (CEEMD) method for nonstationary and nonlinear vibration signals, and the experiment demonstrated that the CEEMD algorithm has good adaptive capability for unstable signals and can effectively extract fault features. Although the above algorithms achieve good results in noise reduction of vibration signal, they also have the following problems: The difficulty of solving the endpoint effect and mode mixing problems of EMD in decomposing vibration signals and the addition of Gaussian white noise to the EEMD decomposition result in a high computational effort and a tendency to decompose spurious IMF components. Although CEEMD solves the endpoint effect and mode mixing problem, there are differences in the number of IMF components generated during decomposition, leading to errors in ensemble averaging.

However, in 2014, Dragomiretskiy et al. [14] proposed the variational mode decomposition (VMD) method, which has better processing effect for strong noise and interference signal processing. The accuracy of the VMD of a vibration signal depends on the decomposition parameter K and the penalty factor $\alpha$. There are different methods for finding the optimal number of layers K and the penalty parameter $\alpha$ of the VMD. For example, Fu et al. [15] used a central frequency observation method to determine the value of the predefined decomposition level K. Yan et al. [16] used solving for the spectral centroid of each IMF component to determine the value of K. Zhan et al. [17] used changes in scattering entropy to determine the optimal K value for the VMD. Zhang et al. [18] used a genetic algorithm combined with nonlinear programming to solve for the VMD parameters. All of the above methods can solve the problem of VMD parameters well, but they all have one-sidedness. Therefore, a better optimization method is sought for noise reduction of the original signal so that the data used for diagnosis can better characterize the fault. Using the noise-reduced signal as sample data for fault diagnosis can better improve the accuracy of fault diagnosis.

With the in-depth research of deep learning theory, more and more scholars have used the theory of deep learning on fault diagnosis, making deep learning an effective means of fault diagnosis [19–21]. The advantage of deep learning methods over traditional machine learning is that they can automatically learn features from raw vibration data, solving the disadvantage of requiring manual extraction of fault features and making deep learning much more accurate for fault diagnosis. Therefore, deep learning fault diagnosis methods based on vibration signals are widely used for all types of mechanical equipment. He et al. [22] proposed a method that uses a combination of vibration signal analysis and deep learning to form a deep learning structure embedded with a time-synchronous resampling mechanism for solving early bearing fault diagnosis. Xu et al. [23] studied a hybrid deep learning model that substantially improved the accuracy of bearing fault diagnosis. Bai et al. [24] used a stacked sparse autoencoder for fault feature dimension reduction and a support vector machine for the diesel engine fault diagnosis method with good results and engineering application value. Li et al. [25] proposed the use of deep learning and multimodal feature fusion approaches to build models for fault diagnosis. Shen et al. [26] used a multilabel convolutional neural network deep learning method to learn relevant features in vibration signals for fault diagnosis, with higher diagnostic accuracy than conventional methods.

Although the above methods can solve the gearbox fault diagnosis problem to a certain extent, there are still the following problems:

1.  The gearbox operating environment is harsh, the weak early fault signal is seriously affected by noise, and fault information is disturbed or masked, making it difficult to reveal fault characteristic information.
2.  Deep learning requires a large amount of labelled data to support it, but in practical engineering applications, the fault states exist for a short time, and large amounts of fault data are difficult to obtain in a short time. To obtain a sufficient amount of data, the equipment needs to fail several times and be in a state of failure for a long time. Once enough data have been collected, a deep learning model with robustness still needs to spend more time on training. These make deep learning methods have major limitations in practical engineering applications.

With the advent of transfer learning, it can effectively solve the problem of fault diagnosis in deep learning, which requires huge amounts of labelled data. Using a transfer learning approach, there is no need to retrain the model, and only a small number of labelled samples are needed to fine-tune the model parameters to achieve good diagnostic results. Yu et al. [27] combined wavelet packet transform and multicore maximum mean square difference to perform deep transfer diagnosis of bearing faults using residual networks (ResNet), which can perform diagnosis and suppress noise effects well. Bai et al. [28] proposed a fault diagnosis method based on transfer learning with optimized variational modal decomposition and deep residual networks, which is effective for noise reduction and fault diagnosis of diesel engines. Su et al. [29] extended convolutional deep belief networks to extract the transportable features from the raw vibration data and used dynamic multilayer perceptron for fault classification, which were experimentally shown to have good classification accuracy for bearing variable condition problems. Luo et al. [30] used the sparse term divergence in the original stacked autoencoder to replace it with a convolutional shortcut to solve the gradient disappearance problem in deep transfer learning and improve feature extraction, which is used for rolling bearing diagnosis with more superior results.

To address these issues, a method combined with the whale optimization algorithm (WOA), VMD, and deep transfer learning is proposed for the fault diagnosis gearboxes. First, the WOA is used to find the optimal decomposition parameters ($K$ and $\alpha$), and the correlation coefficient is used to determine the IMF components and thus select them for signal reconstruction to achieve noise reduction. In the second step, a continuous wavelet transform (CWT) method is used to convert the reconstructed signal into a two-dimensional time–frequency map, which forms the dataset for fault diagnosis. Finally, the AlexNet network is used as the transfer model; after pretraining and fine-tuning, the AlexNet transfer learning (AlexNet-TL) network model is generated to classify the generated 2D time–frequency maps. Experiments have shown that this method can identify fault types quickly and with a higher accuracy.

The main contributions and innovations of this paper are as follows:

1.  For the problem of difficult processing of nonlinear nonsmooth vibration signals, weak fault signals under complex conditions, and difficult extraction of fault features, this paper proposes a WOA-VMD method of signal noise reduction. In complex environments and in more intrusive conditions, the use of this method allows the effects of noise to be well removed, making the fault signature signal more visible.
2.  This paper uses the method of continuous wavelets to turn a one-dimensional vibration signal into a two-dimensional time–frequency image. The good effect of using deep learning on image feature extraction in two dimensions can avoid the blindness of manual extraction of fault features in traditional machine learning, and can effectively improve fault diagnosis accuracy.
3.  This paper uses the fault diagnosis method of model migration. The large amount of data available on ImageNet can be used to train a stable diagnostic model. With the help of model transfer, the need for labelled samples and the reliance on expert

experience can be greatly reduced, making the diagnostic approach more general and generalizable.

The other sections of this paper are detailed as follows: Section 2 describes the noise reduction method of WOA-VMD; Section 3 presents the basic theory of CWT-based time–frequency transformed image generation and deep transfer learning; Section 4 describes in detail the fault diagnosis method steps for WOA-VMD and transfer learning; Section 5 is devoted to experiments and the comparative validation of the proposed fault diagnosis methods; Section 6 presents the conclusions.

## 2. Methodological Theory of Vibration Signal Denoising Based on WOA-VMD

### *2.1. Whale Optimization Algorithm*

The WOA is a heuristic optimization algorithm proposed by Mirjalili and Lewis in 2016 [31], inspired by humpback whale predation behavior. By simulating the bubble net hunting method of humpback whales, a mathematical model is established, which can be used to optimize complex problems. The WOA is divided into two main parts: the hunting phases and exploration phases.

### 2.1.1. Hunting Phase

Whales target the location of their prey. There is a 50/50 chance of whether whales use the constriction mechanism of the bubble net to hunt, or the spiral renewal to get closer to their prey. Mathematical modelling based on this type of hunting is shown in Equation (1):

$$X_{j+1} = \begin{cases} X_j - A \times D & p < 0.5 \\ D' e^{bl} \cos(2\pi l) + X_j^* & p \geq 0.5 \end{cases} \tag{1}$$

where $X^*$ is the position vector of the current optimal solution, $X$ is the current position vector, $D$ is the distance between the humpback whale and the target prey, $D' = \left| X_j^* - X_j \right|$ is the distance between the $i$-th humpback whale and its prey, and the value range of $l$ is [–1, 1]. $p$ is a random number between [0, 1]; $b$ is the constant that determines the shape of the logarithmic helix.

### 2.1.2. Exploration Phase

Humpback whales can randomly search for other prey based on their current location, simulating this behavior by changing the size of $A$. $A$ can take a random value greater than 1 or less than −1 to force the humpback whale away from existing prey to find more suitable prey. The ability to enhance the algorithm's global search with variations of $A$. The mathematical model for searching for prey is shown below:

$$X_{j+1} = X_{rand} - A \times D \tag{2}$$

$$D = \left| C \times X_{rand} - X_j \right| \tag{3}$$

where $X_{rand}$ is a random position vector selected from the existing population.

### *2.2. Variational Mode Decomposition*

The VMD is a solving process of a variational problem. The original signal $f(x)$ is decomposed into $K$ modal functions, and the sum of all modal functions $u_k(t)$ is equal to the original signal as a condition of constraint. It is also guaranteed that the decomposition sequence is the minimum sum of the bandwidths of the modal components of a finite

bandwidth with a central frequency. Therefore, the VMD constrained variational model equation is shown below [32,33]:

$$
\begin{cases}
\min\limits_{\{u_k\},\{\omega_k\}} \left\{ \sum\limits_k \left\| \partial_t[(\delta(t) + \frac{j}{\pi t}) \times u(t)]e^{-j\omega t} \right\|_2^2 \right\} \\
s.t. \sum\limits_k u_k = f(t)
\end{cases}
\tag{4}
$$

where $\delta$ is the Dirac distribution, and $\times$ is the convolution.

To transform the constrained problem in the above equation into an unconstrained variational problem, a quadratic penalty factor and a Lagrangian function are used to introduce the solution process. The quadratic penalty term $a$ and the Lagrange multiplier $\check{}(t)$ are introduced. The extended Lagrangian expression is shown below:

$$
L(\{u_k\},\{\omega_k\},\check{}) = \alpha \sum\limits_k \left\| \partial_t[(\delta(t) + \frac{j}{\pi t}) \times u_k(t)]e^{-j\omega_k t} \right\|^2 + \left\| f(t) - \sum\limits_k u_k(t) \right\|^2 + \left\langle \check{}(t), f(t) - \sum\limits_k u_k(t) \right\rangle
\tag{5}
$$

where $\alpha$ is the penalty factor, and $\check{}$ is the Lagrange multiplier.

The decomposition process of the VMD is as follows:

1. Initialization of $u_k$, $\omega_k$, $\check{}$, and, $n = 0, k = 0$;
2. Starting with $n = n + 1$, iterative computation;
3. Update the modal components $\hat{u}_k^{n+1}$ and the central frequency $\omega_k^{n+1}$ according to the current $u_k$ and $\omega_k$;

$$
\hat{u}_k^{n+1}(\omega) = \frac{\hat{f}(\omega) - \sum\limits_{i \neq k}^{k} \hat{u}_i(\omega) + \hat{\check{}}(\omega)/2}{1 + 2\alpha(\omega - \omega_k)^2}
\tag{6}
$$

$$
\omega_k^{n+1} = \frac{\int_0^\infty \omega \left| \hat{u}(\omega) \right|^2 d\omega}{\int_0^\infty |u_k(\omega)|^2 d\omega}
\tag{7}
$$

4. Update the Lagrange multiplier $\check{}$;

$$
\check{}^{n+1} \leftarrow \check{}^n(\omega) + \tau[\hat{f}(\omega) - \sum\limits_k \hat{u}^{n+1}(\omega)]
\tag{8}
$$

5. Judgement of the end of decomposition.

Given a judgement accuracy of $e > 0$, the decomposition ends when the iteration reaches Equation (9) less than $e$.

$$
\sum\limits_{k=1}^{K} \left( \left\| \hat{u}_k^{n+1} - \hat{u}_k^n \right\|_2^2 / \left\| \hat{u}_k^n \right\|_2^2 \right) < e
\tag{9}
$$

*2.3. Principle and Function of the WOA-VMD*

2.3.1. WOA-VMD Optimization Process and Role

The steps to solve for the optimal parameters of the VMD using the WOA are as follows:

1. The whale population size, maximum number of iterations, spatial dimension, and initial population location were initialized, and the VMD parameters $K$ and $a$ range were set before the WOA began.
2. The VMD of the collected raw vibration signals using the whale's position vector, using the sample entropy as the individual fitness of the initial population.

3. The sample entropy of the current position of the whale is derived after each decomposition until the minimum sample entropy corresponding to the position of the whale appears, that is, the position of the best individual in the current group is obtained to update the spatial position of the individuals in the current group.
4. The position vector of the best individual whale is output, that is, the combination of the decomposition parameters of the VMD is obtained;
5. The role of the WOA-VMD in the overall fault diagnosis is shown in Figure 1.

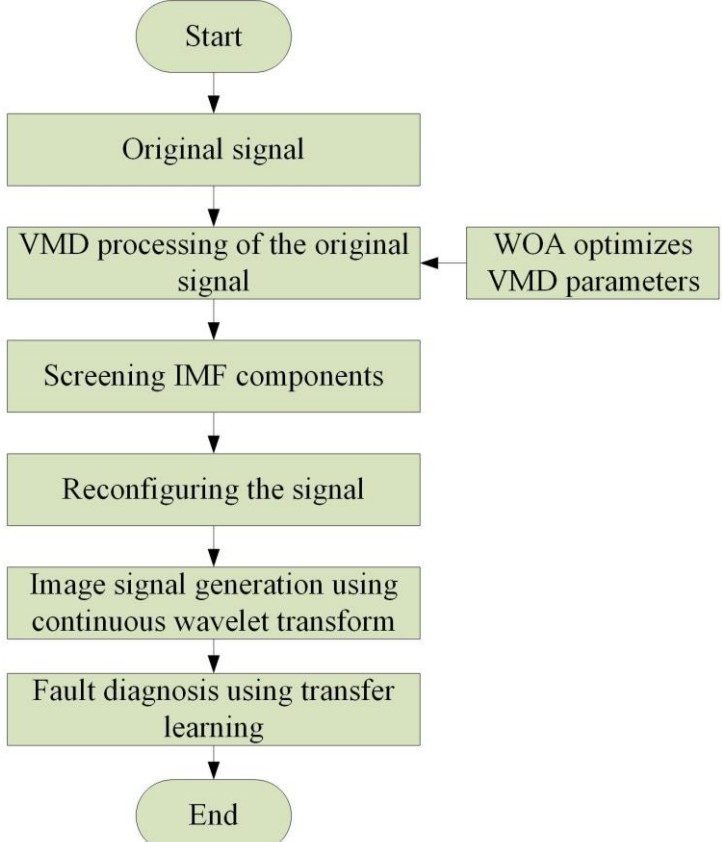

**Figure 1.** The role of WOA-VMD in fault diagnosis.

### 2.3.2. Sample Entropy

Sample entropy [34] has the advantages of being resistant to interference and noise, and is often used in the field of mechanical signal analysis and fault diagnosis. In the VMD, the sample entropy is utilized as the fitness function. The lower the entropy of the sample, the more regular the distribution of the time series, indicating that the IMF obtained by VMD processing contains more valid information. In contrast, the closer the time series is to a random distribution, the more noise components are in the IMF. Therefore, when the sample entropy is smallest, the corresponding parameter is optimal.

### 2.3.3. Evaluation Indicators for Signal Noise Reduction

In order to better verify the noise reduction effect of WOA-VMD, signal-to-noise ratio (SNR) and root mean square error (RMSE) are used as the evaluation index of the noise reduction effect. The SNR is a metric used to compare the desired signal strength with the strength of the background noise. A higher SNR indicates better noise reduction. The RMSE is used to judge the difference between the original signal and the noise reduction

signal, indicating the degree of dispersion of the signal. The smaller the RMSE, the better the noise reduction effect. The calculation formula is shown in Equations (10) and (11):

$$SNR = 10\log\left(\frac{\sum\limits_{i=1}^{n} f(n)}{\sum\limits_{i=1}^{n} [f_d(n) - f(n)]^2}\right) \tag{10}$$

$$RMSE = \sqrt{\frac{1}{n}\sum_{i=1}^{n} (f_d(n) - f(n))^2} \tag{11}$$

where $n$ is the number of samples, $f(n)$ is the original signal, and $f_d(n)$ is the noise reduction signal.

## 3. Time–Frequency Image Generation and Deep Transfer Learning Model Building

### 3.1. Time–Frequency Image Generation

In fault diagnosis, the continuous wavelet transform can convert a one-dimensional vibration signal into a two-dimensional spectral image, allowing the fault signal to be viewed in both the time and frequency domains. Time–frequency diagrams give a good indication of nonstationary signal characteristics and are effective in faulty signal processing. Therefore, the continuous wavelet transform is used in this paper to convert the vibration signal into a two-dimensional time–frequency image [35,36].

### 3.1.1. The Concept of Wavelets

When the functions $\psi(t) \in L^1(R) \cap L^2(R)$ and $\overset{\wedge}{\psi}(0) = 0$, that is, $\int_{-\infty}^{+\infty}\psi(t)dt = 0$, are generated by translation and telescoping the family of functions:

$$\psi_{a,b}(t) = |a|^{-1/2}\psi\left(\frac{t-b}{a}\right), a, b \in R, a \neq 0 \tag{12}$$

where $\psi(t)$ is the base or mother wavelet, $a$ is the scaling factor (also called the scale factor), and $b$ is the translation factor. $\psi_{a,b}(t)$ is the continuous wavelet generated by the base wavelet $\psi(t)$.

### 3.1.2. Continuous Wavelet Transform

For an $\psi_{a,b}(t)$ wavelet function, the wavelet transform is defined for any non-$f(t) \in L^2(R)$ by:

$$Wf(a,b) = |a|^{-1/2}\int_{-\infty}^{+\infty} f(t)\psi^*\left(\frac{t-b}{a}\right)dt \tag{13}$$

The above equation is called the continuous wavelet transform (CWT), where $f(t)$ is the original vibration signal. The continuous wavelet transform transforms the original vibration signal into a two-dimensional time–frequency map signal, as shown in Figure 2.

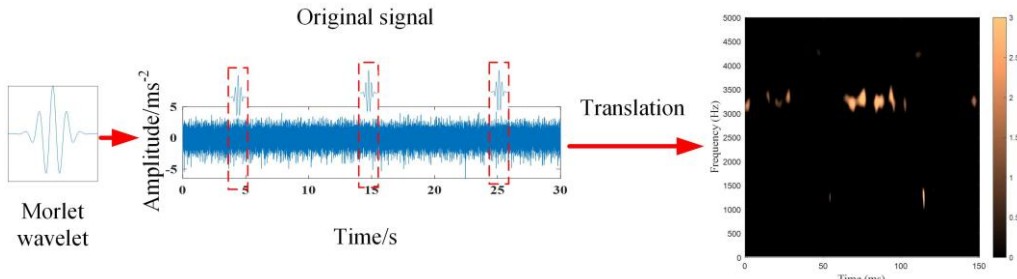

**Figure 2.** Two−dimensional time−frequency image generated by continuous wavelet transform.

### 3.2. Construction of a Deep Neural Network Transfer Model

### 3.2.1. Transfer Learning

Deep learning models for fault diagnosis require a large amount of labelled data to train the model, the larger the amount of data, the more stable the model and the higher the accuracy of the fault diagnosis. The reality is that fault-based tagging data are difficult to obtain, resulting in poor diagnostic accuracy. However, migration learning solves this problem very well. Transfer learning is the migration of a model trained in the source domain to the target domain using a model that has some of the same parameters when used in the source and target domains. Therefore, only a small number of parameters need to be fine-tuned for the target domain in order to carry out fault diagnosis. Figure 3 shows the principle of the deep migration model. In this paper, we choose the AlexNet network model [37] as the transfer target. The AlexNet network model won the championship in image recognition in 2012 and has outstanding image recognition ability. The AlexNet network model is relatively simple, and as a transfer model, there are fewer parameters to adjust, and after fine-tuning the model parameters, a new deep transfer network model, AlexNet-TL, can be generated. Inputting data from the target domain into the AlexNet-TL model for fault diagnosis and diagnosis achieves the desired results. This will save a lot of time and effort compared with the approach of collecting fault data from scratch and training a new deep learning model.

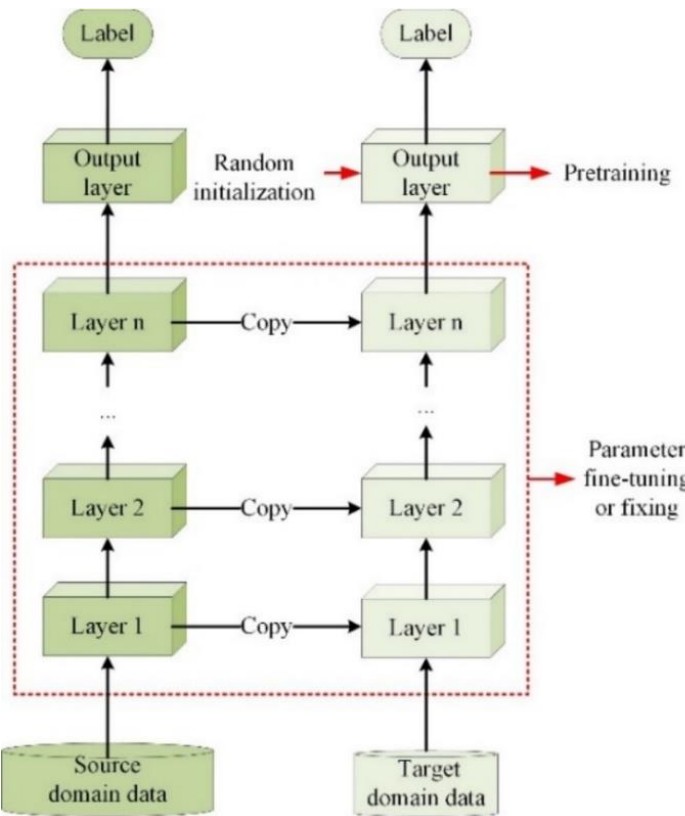

**Figure 3.** Schematic diagram of deep transfer learning.

### 3.2.2. Models for AlexNet Neural Networks

The AlexNet network has five convolutional layers, three maximum pooling layers, and three fully connected layers, as shown in Figure 4.

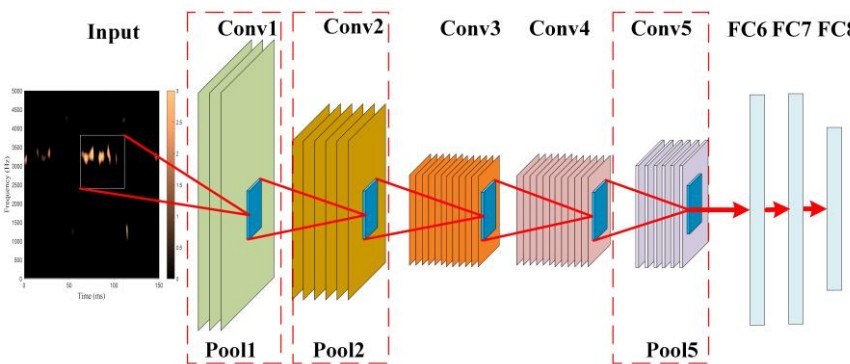

**Figure 4.** AlexNet network model.

1.  Convolutional layers

The role of the convolution layer is to extract the image features of the input image. The input data are computed using a convolution kernel to obtain the output data, which form the output feature map for the next layer. The convolution process is shown in Figure 5.

2.  The formula for the convolution operation is as follows:

$$x_j^n = f\left(\sum_{i \in N_j} x_i^{n-1} * \omega_{ij}^n + b_j^n\right) \tag{14}$$

where $x_j^n$ is the output layer data, $x_i^{n-1}$ is the input layer data, $N_j$ is the convolution region, $n$ is the nth convolution layer, $i$ is the input feature number, $j$ is the output feature number, $\omega_{ij}^n$ is the weight factor, $b_j^n$ is the bias parameter, and $f(\cdot)$ is the activation function. The maximum pooling operation for images is shown in Figure 6.

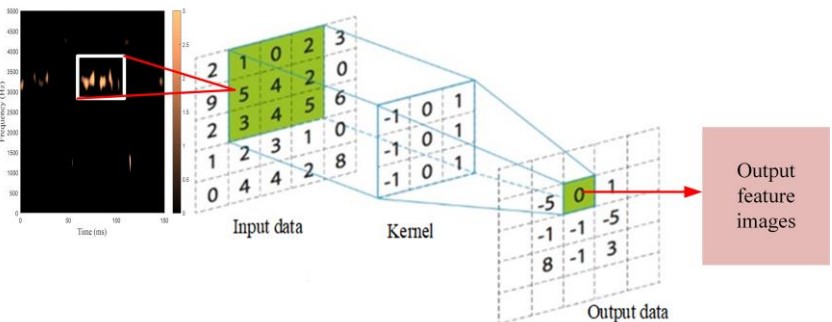

**Figure 5.** Schematic diagram of the convolution process.

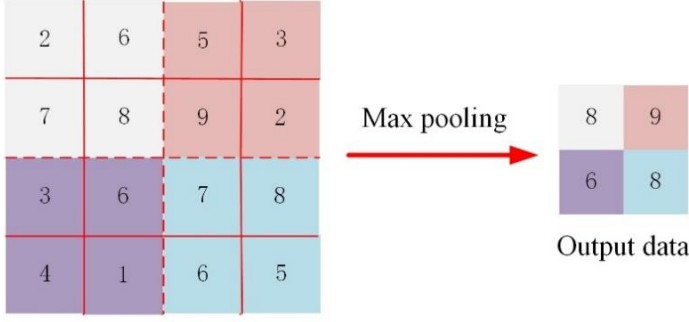

**Figure 6.** Schematic diagram of the max pooling process.

3. Pooling layers

The maximum pooling layer is adopted in the AlexNet network. The main reason for this is that taking maximum pooling allows the image features to be downscaled, and the result of the process avoids the overfitting phenomenon. The maximum pooling operation for images is shown in Figure 6.

The size of the pooling layer is calculated by matrix sliding. The calculation formula is shown below:

$$x_i^m = g(\smile^m * down(x_i^{m-1}) + b^m) \tag{15}$$

where $x_i^m$ is the output data, $x_i^{m-1}$ is the input data, $down(\cdot)$ is the pooling function, $\smile^m$ is the weighting factor, and $b^m$ is the bias parameter.

4. Fully connected layer

The role of the fully connected layer is to reduce the dimensionality of the data in order to prevent the loss of important data feature information in the image caused by the data going directly from the convolution layer to the output layer. The full connection is located at the end of the AlexNet network and connects the output layer, which is classified by softmax. The formula for its calculation is shown below:

$$x^l = Softmax(\omega^l * x^{l-1} + b^l) \tag{16}$$

where $x^l$ is the fully connected layers output data, $x^{l-1}$ is the fully connected layers' input data, $\omega^l$ is the weighting factor, $b^l$ is the bias parameter, and $l$ is the $l$th layer network.

From the AlexNet network structure, it can be seen that after the 2D image is processed by five convolutional layers and three maximum pooling layers, the features of the image are further extracted, and the AlexNet network obtains deeper features of the 2D image. After the fully connected layers' processing and then classification, a better classification result can be obtained.

## 4. Gearbox Fault Diagnosis Based on WOA-VMD and Deep Transfer Learning

The flow of the gear fault diagnosis method proposed in this paper, represented in Figure 7, is divided into the following four key processes:

**Step 1. Vibration signal noise reduction and reconstruction:** Using the experimental platform to collect data for different working conditions, the WOA-VMD method is used to decompose the original signal and solve for the sample entropy corresponding to each IMF component. When the sample entropy is smallest, the corresponding K and $\alpha$ values are the optimal parameters. The correlation coefficient between the solved IMF components and the original signal is then used to further determine the relationship between the IMF components and the original signal, and the IMF components with high correlation coefficients are then selected and added together for reconstruction to obtain the denoised signal.

**Step 2. 2D time–frequency plot conversion and dataset generation:** Using the signal reconstructed after denoising in the previous step as the input condition, a CWT method is used to convert the one-dimensional vibration signal into a two-dimensional time–frequency signal dataset.

**Step 3. Generation of AlexNet-TL network models:** The AlexNet network model was used as the transfer target. In the fine-tuning of the model parameters, the parameters of the first five convolutional layers are kept frozen, and the parameters of the last three fully connected layers are fine-tuned, and the training data from the generated dataset are fed into the pretrained model for training. A new network model, named AlexNet-TL, was generated after fine-tuning the parameters of the AlexNet network.

**Step 4. Fault pattern recognition with AlexNet-TL:** The test set and validation set data for the four operating conditions are imported into the AlexNet-TL network model to obtain the results of the fault diagnosis.

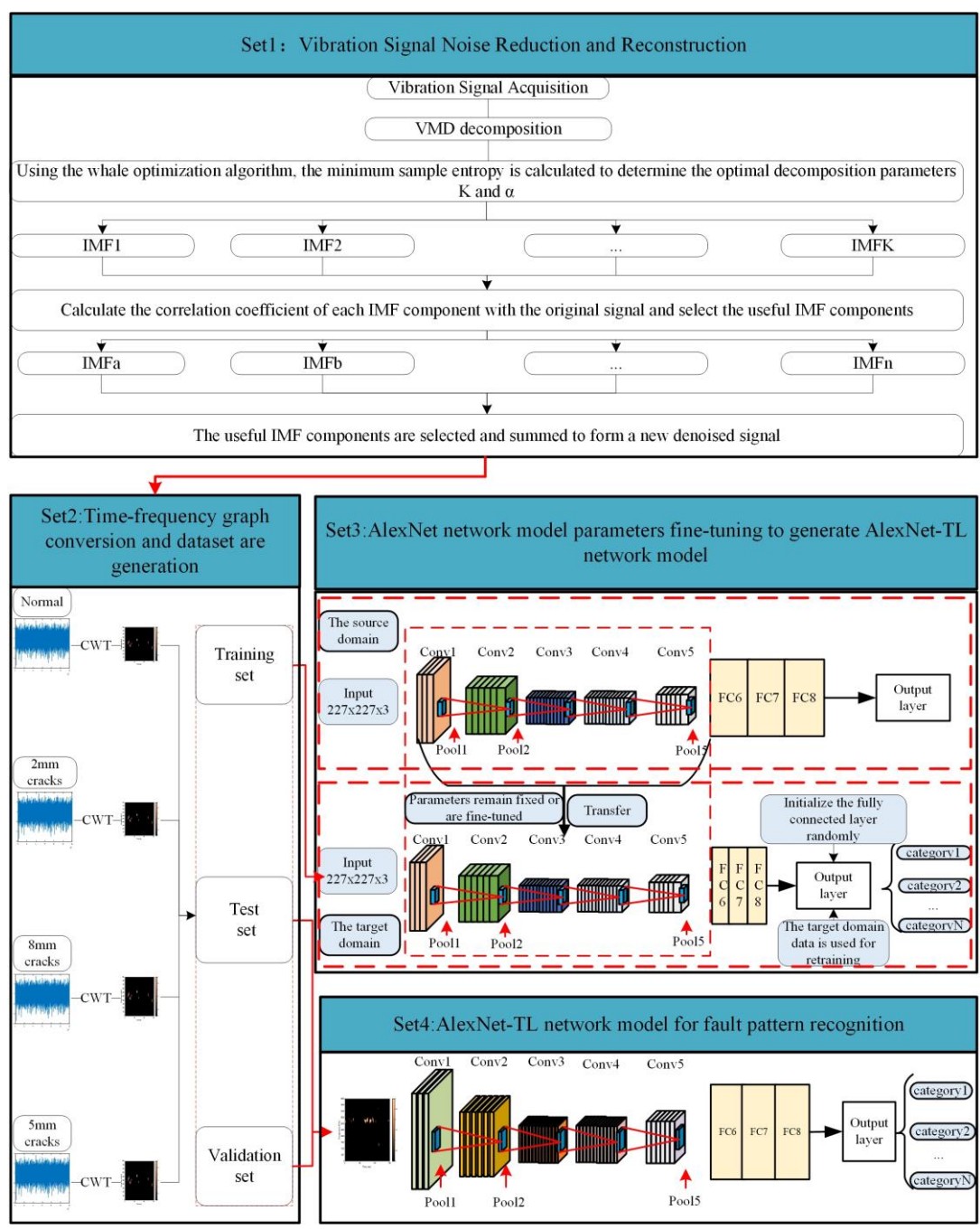

**Figure 7.** Flowchart for gearbox fault diagnosis based on WOA-VMD and deep transfer learning.

## 5. Experimental Process and Method Validation

In order to verify the feasibility of the theoretical approach proposed in this paper, experiments were carried out using a laboratory gearbox experimental platform with pre-set faults.

### 5.1. Introduction to the Experimental Platform

The preset experiments were carried out on a JZQ175 speed reducer manufactured by the General Machinery Factory in Hejian City of Hebei Province, China. The vibration sensor model used in the experiments is IEPE general purpose no. 14100. The structure of the experimental platform is shown in Figure 8. The experimental platform mainly consists

of a base, a three-phase asynchronous motor, an electromagnetic speed control motor, a planetary gear reducer, and a magnetic powder brake.

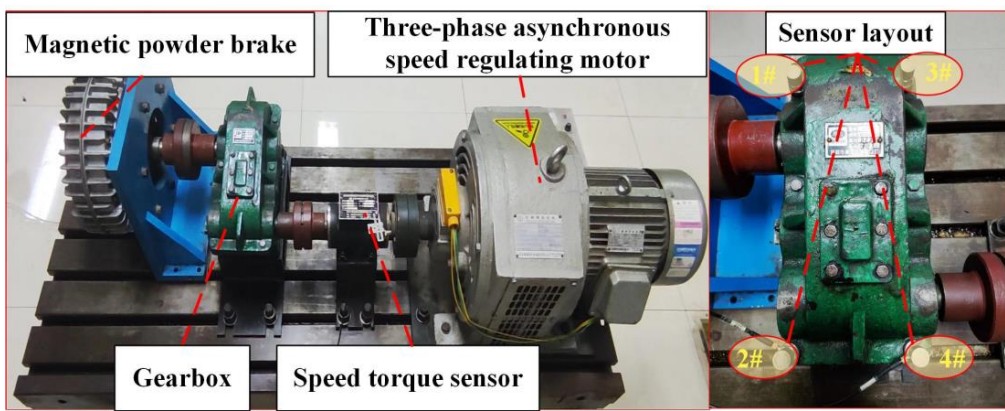

**Figure 8.** Gear reducer test stand and sensor layout.

This paper focuses on the early failure of gears with cracks. Using the experimental platform, the vibration signals of the gearbox gears were tested under normal 2 mm crack, 5 mm crack, and 8 mm crack, as shown in Table 1. The cracking of the gears in the test was set up using the CNC lathe, and the fault presets are shown in Figure 9.

**Table 1.** Fault presetting modes for gearheads.

| Serial Number | Working Condition Labels | Fault Status |
|---|---|---|
| 1 | S1 | Normal |
| 2 | S2 | 2 mm crack |
| 3 | S3 | 5 mm crack |
| 4 | S4 | 8 mm crack |

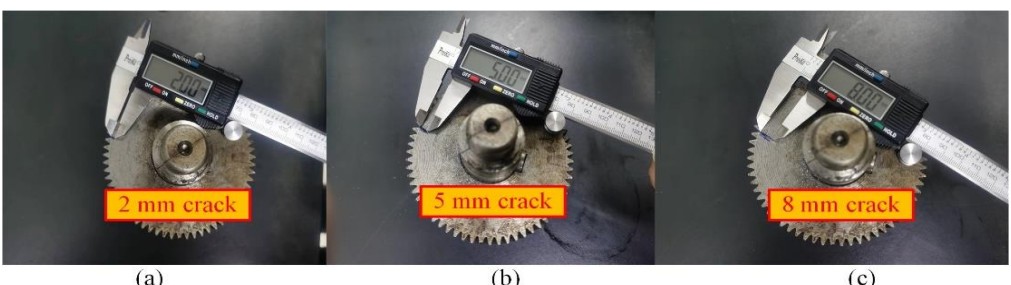

**Figure 9.** Fault preset states: (**a**) 2 mm crack, (**b**) 5 mm crack, (**c**) 8 mm crack.

### 5.2. Vibration Signal Acquisition and Preprocessing

5.2.1. Vibration Signal Acquisition Scheme

Signal acquisition was carried out under stable operating conditions on the experimental platform. At this point, the motor input speed was 800 r/min, and five sets of data were collected for each operating condition. The sampling frequency of the vibration sensor was 20 kHz, and the data acquisition time for each group was 6 s, with the next group of data acquired at an interval of 5 s. The data collection for the four different operating conditions is shown in Table 2. Figure 10 shows the time domain waveforms of the vibration signals for the four operating conditions of the gear.

**Table 2.** Experimental data acquisition scheme.

| Labels | Sample Frequency | Sampling Time | Sampling Interval | Data Groups | Input Speed | Number of Sensors |
|---|---|---|---|---|---|---|
| S1 | 20 kHz | 6 s | 5 s | 5 | 800 r/min | 4 |
| S2 | 20 kHz | 6 s | 5 s | 5 | 800 r/min | 4 |
| S3 | 20 kHz | 6 s | 5 s | 5 | 800 r/min | 4 |
| S4 | 20 kHz | 6 s | 5 s | 5 | 800 r/min | 4 |

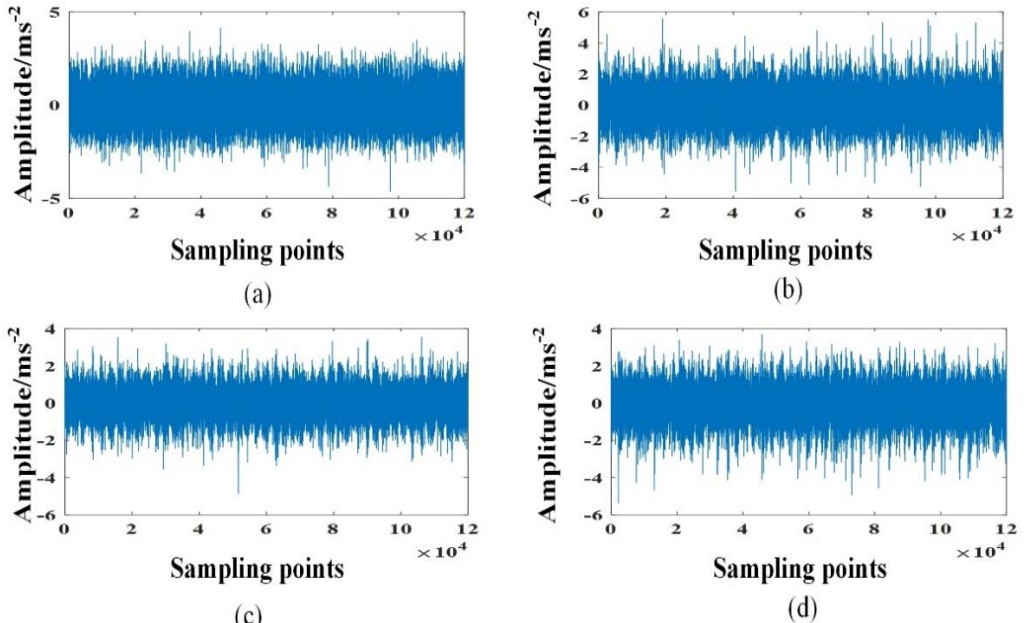

**Figure 10.** Vibration waveforms of gear reducer under four different states: (**a**) S1 state, (**b**) S2 state, (**c**) S3 status, (**d**) S4 state.

5.2.2. Signal Preprocessing

The experimental data collected by Sensor 2 are used in this paper for experimental verification. The collected raw signals are processed using WOA−VMD to find the values of the optimal decomposition parameters *K* and *a* using the sample entropy as the fitness function, and the results are shown in Table 3. Considering the same treatment method, the S4 working condition is used as an example. At this point, the smallest sample entropy corresponding to a *K* value of 7 and an *a* value of 2000 is obtained, and the time and frequency domain plots after the VMD are shown in Figure 11.

**Table 3.** The corresponding *K* and α of WOA−VMD under different fault states and minimum sample entropy.

| Labels | Sample Entropy Minimum | K | α |
|---|---|---|---|
| S1 | 0.892 | 8 | 2000 |
| S2 | 0.693 | 8 | 1980 |
| S3 | 0.598 | 9 | 1920 |
| S4 | 0.893 | 7 | 2000 |

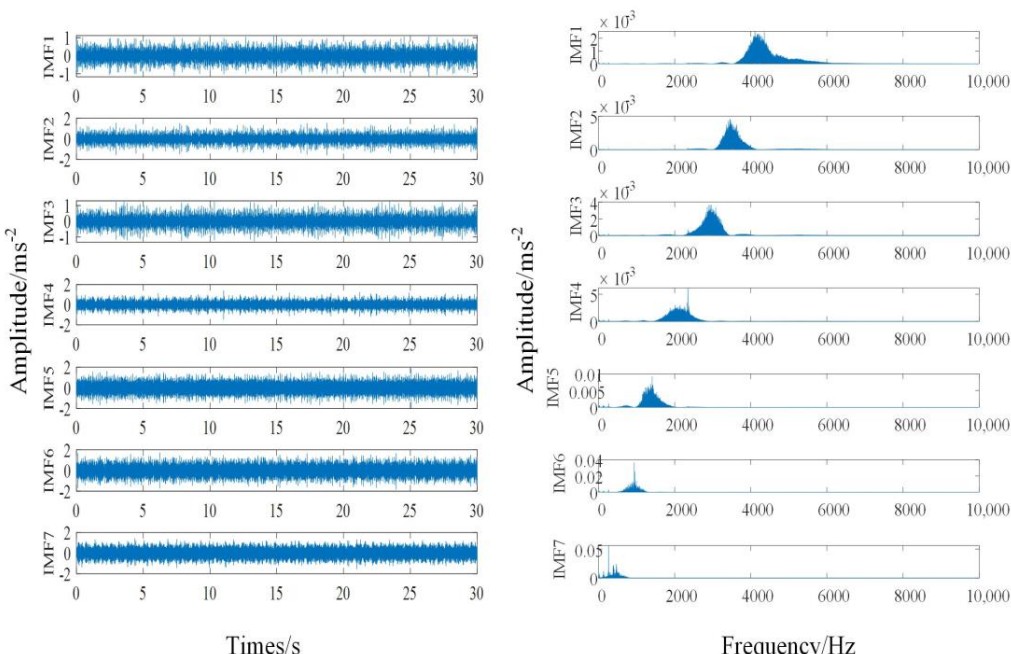

**Figure 11.** Time–frequency diagram of VMD: (**a**) time domain plots for each IMF component; (**b**) frequency domain plots for each IMF component.

To optimize the signal even further and to remove noise from the signal, a correlation coefficient [38] was introduced for this purpose to judge the IMF components from the VMD. The calculation formula is shown below:

$$\rho_{XY} = \frac{1}{n} \frac{\sum\limits_{i=1}^{n} (X_i - E(X))(Y_i - E(Y))}{\sqrt{\frac{\sum\limits_{i=1}^{n} (X_i - E(X))^2}{n}} \sqrt{\frac{\sum\limits_{i=1}^{n} (Y_i - E(Y))^2}{n}}} \tag{17}$$

where $n$ is the number of samples, $X_i$ is the original signal, and $Y_i$ is the corresponding IMF component; $E(X) = \frac{\sum\limits_{i=1}^{n} X_i}{n}$ I s the mean value of the original signal; $E(Y) = \frac{\sum\limits_{i=1}^{n} Y_i}{n}$ for the IMF component.

After WOA-VMD, the correlation coefficients between the IMF components and their respective original signals for the four operating conditions are shown in Table 4 below.

**Table 4.** Correlation coefficient between IMF component and original signal.

| Labels | IMF1 | IMF2 | IMF3 | IMF4 | IMF5 | IMF6 | IMF7 | IMF8 | IMF9 |
|--------|------|------|------|------|------|------|------|------|------|
| S1 | 0.1731 | 0.2878 | 0.2253 | 0.2767 | 0.3655 | 0.6305 | 0.4508 | 0.4862 | |
| S2 | 0.1521 | 0.1925 | 0.2922 | 0.3908 | 0.4675 | 0.5295 | 0.5356 | 0.4887 | |
| S3 | 0.1911 | 0.2995 | 0.3583 | 0.3425 | 0.3444 | 0.3863 | 0.4512 | 0.4803 | 0.4961 |
| S4 | 0.2627 | 0.3690 | 0.3391 | 0.3439 | 0.5131 | 0.5808 | 0.4981 | | |

According to the principle of correlation coefficient, the higher the value of the correlation coefficient, the stronger the correlation; thus, the IMF components with a higher correlation degree are selected from the table for signal reconstruction. A continuous wavelet approach is taken to the reconstructed signal, which is transformed into a two-dimensional time–frequency map and used as the target domain dataset for transfer learning. The size of the time–frequency image in pixels is set to 227 × 227, and the two-dimensional time–frequency images for the four different operating conditions are shown in Figure 12. As can be seen from Section 5.2.1, the number of points sampled for each operating condition

is 600,000, and a two-dimensional time–frequency image sample is generated at every 3000 points; then the dataset used for the experiment is shown in Table 5.

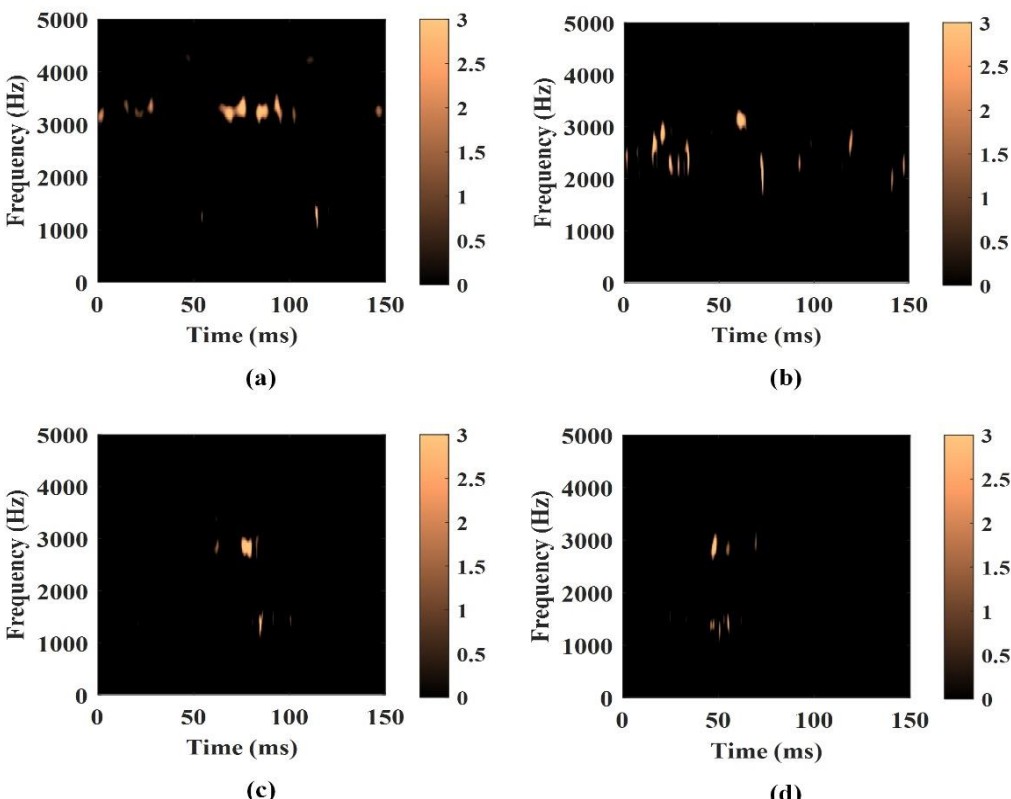

**Figure 12.** Two-dimensional time–frequency images of four different operating conditions: (**a**) S1 state, (**b**) S2 state, (**c**) S3 status, (**d**) S4 state.

**Table 5.** Time–frequency plot datasets for four different fault states after WOA-VMD.

| Labels | Number of Samples | Training Samples | Test Samples | Validation Samples |
|--------|-------------------|------------------|--------------|---------------------|
| S1 | 240 | 180 | 20 | 40 |
| S2 | 240 | 180 | 20 | 40 |
| S3 | 240 | 180 | 20 | 40 |
| S4 | 240 | 180 | 20 | 40 |
| Total | 960 | 720 | 80 | 160 |

*5.3. Determination and Training of AlexNet-TL Model Parameters*

The direct use of the AlexNet network for fault diagnosis is not ideal, and therefore, the AlexNet network needs to be adapted. The main modifications are in the last three fully connected layers of the AlexNet network. The AlexNet network was trained using the training and test samples in Table 5, and then its network model parameters were fine-tuned. The parameters of the AlexNet-TL model after adjustment are shown in Table 6.

To demonstrate the effectiveness of the AlexNet-TL model proposed in this paper for gear fault diagnosis, a feature visualization approach is adopted for validation. The dataset in Table 5 was fed into AlexNet-TL for training, and the t-distributed random neighborhood embedding (t-SNE) method was adopted to visualize the distribution of image features for different working conditions on different layers. The scatter plots of the four different working condition image features on different layers are shown in Figure 13. The figure shows that at pooling layer 1 and pooling layer 2, there is a cross-mixing of 5 mm crack fault characteristics and 8 mm crack failure signatures that are not well separated. As learning progresses, the 5 mm crack fault features and 8 mm crack fault features can be

well separated at the fifth convolutional layer, but fault features of 2 mm cracks and normal states still cross-mixing affects the fault diagnosis accuracy. As the fault feature learning progresses to the seventh fully connected layer, the fault features of the four operating conditions can be perfectly separated. It can thus be shown that the AlexNet-TL model with adjusted parameters in this paper is superior for gear fault diagnosis.

**Table 6.** Model parameters of AlexNet-TL.

| Layer Name | Input Size | Output Size | Activation Function |
|---|---|---|---|
| Input | $227 \times 227 \times 3$ | - | - |
| Conv1 | $227 \times 227 \times 3$ | $55 \times 55 \times 96$ | ReLU |
| Maxpooling1 | $55 \times 55 \times 96$ | $27 \times 27 \times 96$ | - |
| Conv2 | $27 \times 27 \times 96$ | $27 \times 27 \times 256$ | ReLU |
| Maxpooling2 | $27 \times 27 \times 256$ | $13 \times 13 \times 256$ | - |
| Conv3 | $13 \times 13 \times 256$ | $13 \times 13 \times 384$ | ReLU |
| Conv4 | $13 \times 13 \times 384$ | $13 \times 13 \times 384$ | ReLU |
| Conv5 | $13 \times 13 \times 384$ | $13 \times 13 \times 256$ | ReLU |
| Maxpooling5 | $13 \times 13 \times 256$ | $6 \times 6 \times 256$ | - |
| Fully connected6 | $6 \times 6 \times 256$ | 512 | ReLU |
| Fully connected7 | 512 | 256 | ReLU |
| Fully connected8 | 256 | 4 | - |
| Output | - | 4 | - |

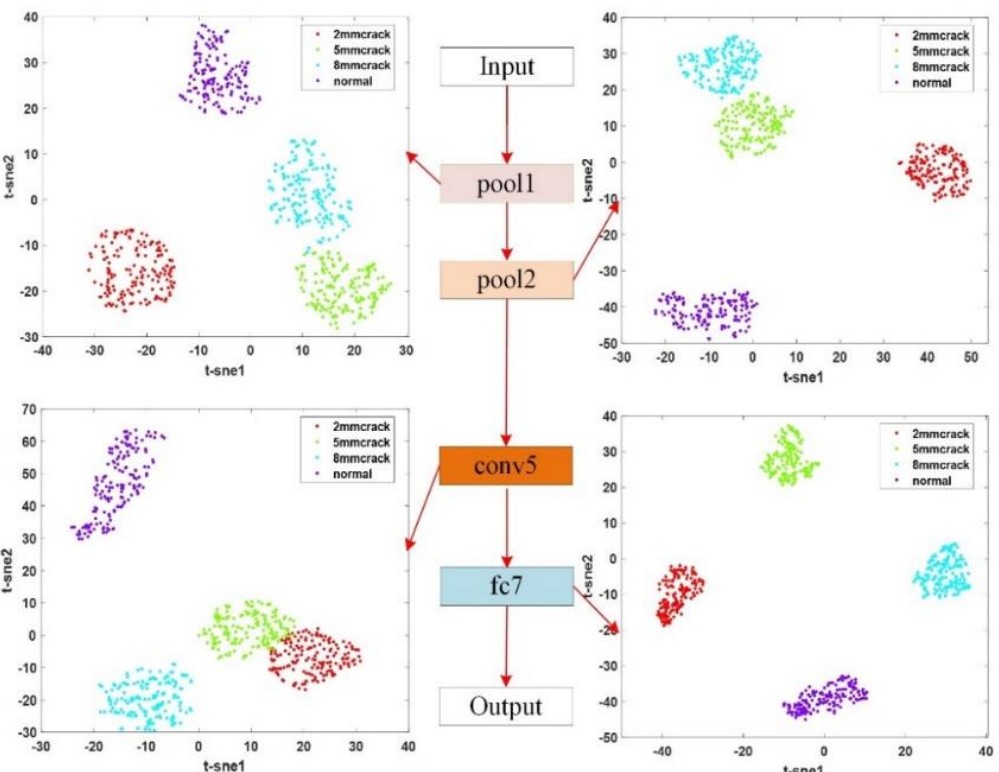

**Figure 13.** Scatter plot of image features on different layers for four different working conditions.

*5.4. Comparative Analysis of the Effectiveness of Different Noise Reduction Methods*

5.4.1. Comparative Analysis of Signal Noise Reduction Effects

To verify the superiority of WOA−VMD noise reduction, its noise reduction effect is compared with EMD, EEMD, CEEMD, VMD, and other methods. The SNR and RMSE were calculated for different operating conditions using different noise reduction methods, and the results are shown in Table 7. The table shows that the SNR of the WOA−VMD noise reduction method adopted in this paper is larger, and the RMSE is smaller than that of several other methods. Combined with the judging criteria in Section 2.3.3, the superiority

of the WOA−VMD noise reduction effect adopted in this paper can be illustrated from these two aspects.

**Table 7.** SNR and RMSE after using different denoising methods.

| Labels | Evaluation Indicators | EMD | EEMD | CEEMD | VMD | WOA-VMD |
|:---:|:---:|:---:|:---:|:---:|:---:|:---:|
| S1 | *SNR* | 3.23 | 4.36 | 5.27 | 7.31 | 7.89 |
|  | *RMSE* | 0.58 | 0.63 | 0.66 | 0.50 | 0.45 |
| S2 | *SNR* | 2.18 | 4.63 | 5.36 | 6.45 | 8.86 |
|  | *RMSE* | 0.66 | 0.59 | 0.56 | 0.46 | 0.39 |
| S3 | *SNR* | 3.42 | 5.34 | 4.23 | 6.77 | 8.34 |
|  | *RMSE* | 0.72 | 0.57 | 0.61 | 0.51 | 0.44 |
| S4 | *SNR* | 2.63 | 5.28 | 3.15 | 6.24 | 8.18 |
|  | *RMSE* | 0.64 | 0.61 | 0.47 | 0.53 | 0.40 |

5.4.2. Comparative Analysis of Fault Diagnosis Results Corresponding to Different Noise Reduction Methods

To better illustrate the effect of noise reduction on the diagnostic results, the noise reduced signal is fed into the AlexNet−TL network for fault diagnosis, and the diagnostic results are used to judge the effect of noise reduction. EMD, EEMD, CEEMD, and WOA-VMD were used to reconstruct the original signal for noise reduction, and the reconstructed signal was converted into a two-dimensional time–frequency image signal by the CWT method. The above time–frequency image signals were used as the target domain dataset and input to AlexNet−TL for fault classification, and the classification results are shown in Table 8.

**Table 8.** Comparative analysis of fault diagnosis results corresponding to different noise reduction methods.

| Labels | Original Signal | EMD | EEMD | CEEMD | VMD | WOA-VMD |
|:---:|:---:|:---:|:---:|:---:|:---:|:---:|
| S1 | 95.0% | 97.5% | 100.0% | 100.0% | 100.0% | 100.0% |
| S2 | 82.5% | 92.5% | 95.0% | 97.5% | 97.5% | 100.0% |
| S3 | 80.0% | 97.5% | 95.0% | 95.0% | 97.5% | 100.0% |
| S4 | 57.5% | 90.0% | 92.5% | 95.0% | 97.5% | 100.0% |
| Accuracy | 78.75% | 93.75% | 95.63% | 96.88% | 98.125% | 100.0% |

Figure 14 shows the results of the new and original signals after reconstruction using WOA−VMD, VMD, CEEMD, EEMD, and EMD noise reduction methods and fault diagnosis using AlexNet−TL. The accuracy rates for fault diagnosis were 100.0%, 98.125%, 96.88%, 95.63%, 93.125%, and 78.75%, respectively. In terms of classification results, the use of noise reduction methods can greatly improve fault classification accuracy. Even the EMD classification with the worst noise reduction achieved an accuracy of 93.125%, much higher than the 78.75% of the original signal. This shows the importance of signal noise reduction in fault diagnosis. The diagnostic accuracy of WOA-VMD was the highest compared with the remaining five noise reduction methods, reaching 100.0%. It can be shown that the signal after noise reduction using WOA-VMD is the most effective for fault diagnosis. Therefore, the noise reduction method chosen in this paper is effective.

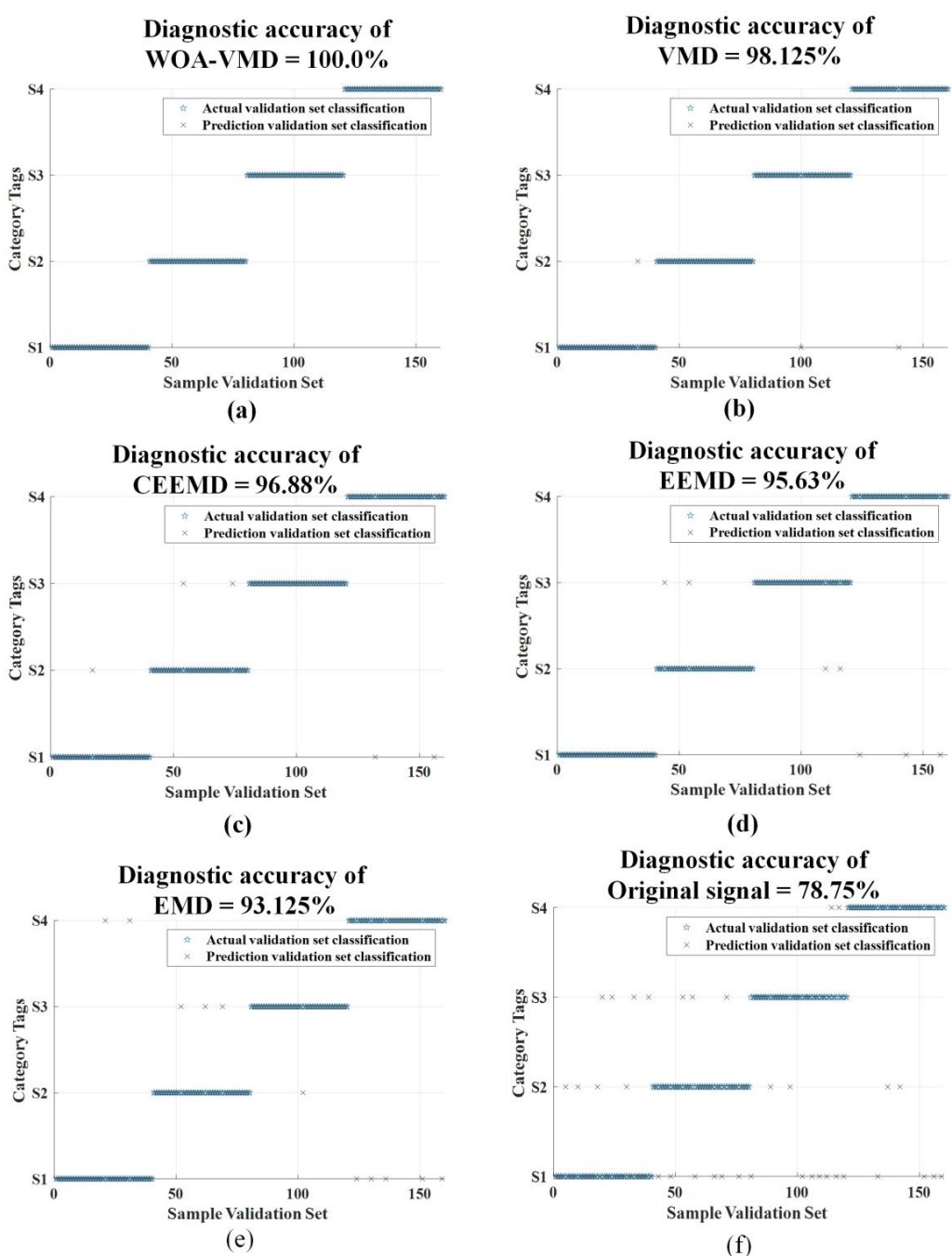

**Figure 14.** Comparative analysis of diagnostic results of different noise reduction methods: (**a**) WOA-VMD = 100.0%; (**b**) VMD = 98.125%; (**c**) CEEMD = 96.88%; (**d**) EEMD = 95.63%; (**e**) EMD = 93.125%; (**f**) original signal = 78.75%.

*5.5. Comparative Analysis of the Diagnostic Effects of Different Neural Network Models*

In order to demonstrate that the diagnostic effect of AlexNet−TL proposed in this paper is better than other networks, the dataset in Table 5 was selected for fault diagnosis using the AlexNet−TL network model proposed in this paper and the GoogLeNet, ResNet18, and SqueezeNet network models. The fault diagnosis results for the four different networks are shown in Figure 15. As you can see from the graph, AlexNet−TL has the highest accuracy rate at 100%. SqueezeNet and ResNet18 also demonstrate their powerful image feature learning capabilities and can achieve 100% recognition accuracy for some working conditions. Although there is still a gap in overall accuracy compared with AlexNet−TL, this is under small sample conditions, and AlexNet−TL has been pretrained

and fine-tuned prior to diagnosis. Overall, the AlexNet−TL neural network demonstrated excellent diagnostic accuracy for gear faults, with a much higher accuracy rate compared with the other three neural network models. The superiority of the proposed diagnostic method can be further illustrated by the comparison of the diagnostic results.

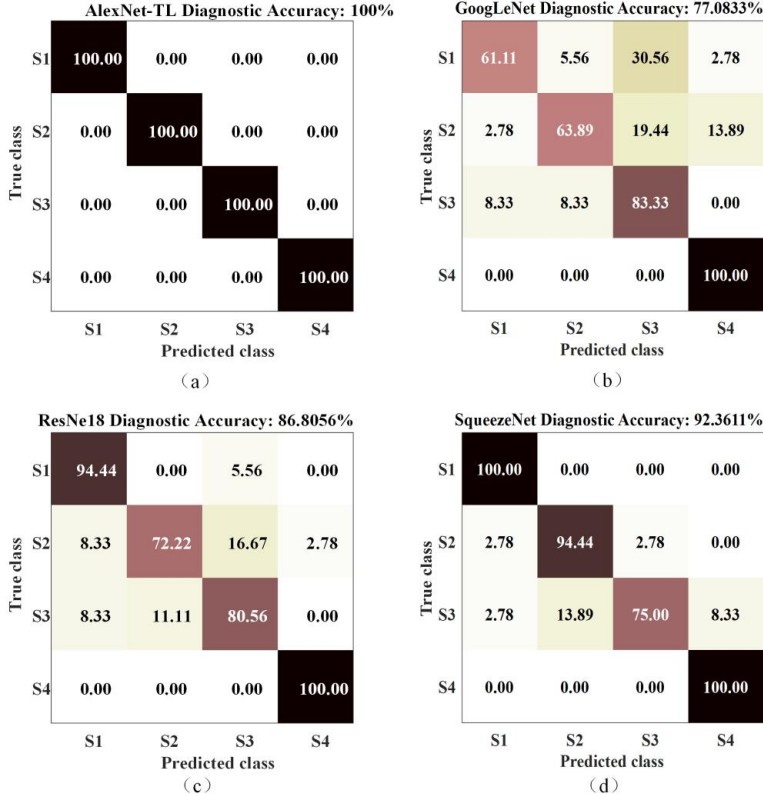

**Figure 15.** Fault diagnosis results of different network models: (**a**) AlexNet−TL = 100.0%; (**b**) GoogLeNet = 77.0833%; (**c**) ResNet18 = 86.8056%; (**d**) SqueezeNet = 92.3611%.

*5.6. Comparative Analysis of the Diagnostic Effects of Different Fault Diagnosis Methods*

To further illustrate the novelty and superiority of the fault diagnosis methods adopted in this paper, the fault diagnosis methods used in this paper are compared with the five fault diagnosis methods mentioned in the introduction section. All diagnostic methods make use of the same experimental data. The experimental data are derived from Section 5.2.1. The five diagnostic methods mentioned in the introduction section are (1) hybrid deep signal processing methods for fault diagnosis, (2) hybrid deep learning models based on CNN and gcForest for fault diagnosis (CNN−gcForest), (3) fault diagnosis method based on superimposed autoencoders and support vector machines (SSAE-SVM), (4) fault diagnostic method based on deep learning technology and multimodel feature fusion (MMFF−FD), and (5) fault diagnostic method based on deep multilabel learning framework called multilabel convolutional neural network (MLCNN). The results of the diagnosis are shown in Table 9.

**Table 9.** Diagnostic results of different fault diagnosis methods.

| Diagnostic Methods | Hybrid Deep Signal Processing | CNN-gcForest | SSAE-SVM | MMFF-FD | MLCNN | AlexNet-TL |
|---|---|---|---|---|---|---|
| S1 | 94.44% | 94.44% | 94.44% | 97.22% | 97.22% | 100.0% |
| S2 | 97.22% | 97.22% | 97.22% | 100.0% | 100.0% | 100.0% |
| S3 | 100.0% | 97.22% | 97.22% | 97.22% | 97.22% | 100.0% |
| S4 | 100.0% | 97.22% | 91.67% | 94.44% | 100.0% | 100.0% |
| Diagnosis Accuracy | 97.92% | 96.53% | 95.14% | 97.22% | 98.61% | 100.0% |

Figure 16 shows the specifics of the diagnostic results of the five fault diagnosis methods and the methods in this paper. As can be seen from the graph, AlexNet-TL's method had the highest accuracy rate of 100.00%. The classification accuracy is the highest compared with the other five diagnostic methods. Of these five methods, SSAE-SVM and MMFF-FD are fault diagnosis methods that use one-dimensional vibration signals to extract fault features before classifying faults. In contrast, the method used in this paper transforms a one-dimensional vibration signal into a two-dimensional time–frequency image, and then uses deep migration learning to extract picture fault features for fault classification. Obviously, the fault characteristics are more prominent in two-dimensional time–frequency space, making the method used in this paper more accurate in diagnosis. The remaining three methods also use deep learning to extract 2D time–frequency image features for fault diagnosis, but are limited by small samples, resulting in gaps in fault classification accuracy. The approach used in the paper draws on the strengths of deep learning and transfer learning. One is the advantage of deep learning's strong capability in image feature extraction. Second, transfer learning has the advantage of being better at solving small sample problems. Thus, the novelty of the diagnostic method proposed in this paper can be demonstrated compared with the other five methods.

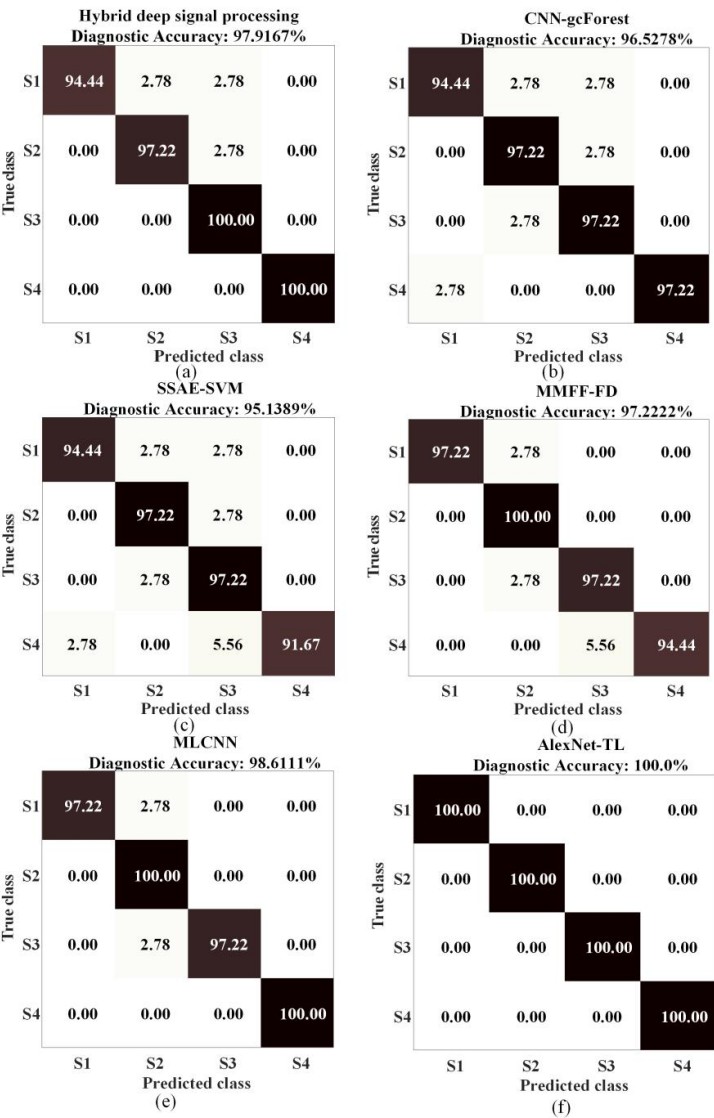

**Figure 16.** Diagnostic results of different diagnostic methods: (**a**) hybrid deep signal processing = 97.92%; (**b**) CNN−gcForest = 96.53%; (**c**) SSAE−SVM = 95.14%; (**d**) MMFF-FD = 97.22%; (**e**) MLCNN = 98.61%; (**f**) AlexNet−TL = 100.0%.

## 6. Conclusions

This paper proposed a fault diagnosis method using a combination of WOA-VMD and AlexNet−TL. The essential findings are as follows:

1. Using the WOA with sample entropy as an adaptation function, the optimal decomposition level K and penalty parameter $\alpha$ can be found quickly to achieve better noise reduction.
2. The CWT method is used to program a one-dimensional vibration signal into a two-dimensional time–frequency image signal, which is combined with the good performance of deep transfer networks for image recognition for fault classification of gear faults.
3. A reliable fault diagnosis method based on deep transfer learning is proposed. The AlexNet network was chosen as the transfer object, and through pretraining and fine-tuning of the model, it has a good recognition effect on early gear faults of gearboxes, solving the problem of small sample constraints in fault diagnosis.

The method proposed in this paper only investigates the single fault of a cracked gear. In practice, however, gears work for long periods in harsh environments, often with multiple faults occurring together. At present, we have not conducted enough research on fault diagnosis methods for multiple fault conditions and will focus on this direction in the next step.

**Author Contributions:** Conceptualization, Z.W., H.B., H.Y. and X.Z.; methodology, Z.W. and H.B.; software, Z.W. and H.B.; data curation, Z.W. and H.Y.; formal analysis, Z.W.; resources, Z.W., C.G. and X.J.; validation, Z.W. and X.Z.; visualization, Z.W.; investigation, Z.W., X.Z. and H.Y.; writing—originaldraft prepa-ration, Z.W.; writing—review and editing. Z.W., H.B., C.G. and X.J.; funding acquisition, C.G. and X.J.; project administration, C.G. and X.J.; supervision, C.G. and X.J. All authors have read and agreed to the published version of the manuscript.

**Funding:** This work is supported by the National Natural Science Foundation of China (grant No.71871220). The support is gratefully acknowledged. The authors would also like to thank thereviewers for their valuable suggestions and comments.

**Data Availability Statement:** The data involved in this article have been presented in the article.

**Conflicts of Interest:** The authors declare no conflict of interest.

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
