# Peer review of "Intelligent Fault Diagnosis Method for Gearboxes Based on Deep Transfer Learning"

_processes, doi:10.3390/pr11010068_

Round 1

Reviewer 1 Report

This paper is very well written. It is clear and very ease to read.

In terms of thechincal comments, this is what I have:

*line 191: could you please justify starting with "n+1"

*line 272: Figure 3: Layer 1 is repeated twice. Ifmy understanding is correct, it should layer 1, 2 through n

*line 462: I suggest using both the 8mm crack and the 2mm crack to better prove the efficiency of your technique.

*In general, I would also suggest emphasizing the fact that ALL components of your technique are essential to achieve the results obtained.

*Please calrify the difference between this paper and reference [33] 

NOW, some formatting issues:

*line 60: "However, in 2014: a comma is missing after 2014

*line 60 to 73: too many ";". Replace them with "." where appropriate - otherwise the paragraph looks lige a very long sentence

*line 97: remove the ";" at the end

*line 145 through 150: It is better to replace "chapter" and "part" by "section"

line 364: Please clarify "fault presetting"

*line 391: "are shown in 11" - I assume you are talking about Figure 11?

Reviewer 2 Report

1-      The novelty and research gaps are unclear. The novelty of this work is not clear; so it needs more justification.

2-      In equations, explain all the parameters. Check all formulas in the document. They are not formatted. Define the used symbols clearly and numerate all equations that appear.

3-      The detailed information on the instruments used must be given including the model, manufacturer, name of company, place where the company located, etc.

4-      It is better to compare the software results with the results of the other studies. Please compare your results and research with the results of other authors and underline what is the scientific novelty in this work.

5-      The Limitations of the proposed study need to be discussed before conclusion.

6-      Author should add separate section regarding future outlook and specific comment point wise based on their study.

7-      English is generally good; I think it needs to be polished further and some typos need to be revised.

Reviewer 3 Report

The abstract is written in an overview manner.
The Introduction provides an adequate overview of works in the given area, summarizing different approaches with their advantages and problematic aspects. He also mentions the key problem of using deep learning methods for the given area, which is that there is not enough data to learn from. Therefore, the authors used a heuristic algorithm in their work.
I recommend that fig. 1 should not contain terms such as whale position but be transformed into the task of diagnosing gearbox faults. Alternatively, to create a similar diagram to be generated for the processing of diagnostic transmission signals, which is partially described in the steps at the beginning of the 4th chapter. If the given function should be performed by fig. 7, in this form it does not appear clear enough.
Formal shortcomings - the equations on page 5 are apparently inserted as fig. and their scaling is not good.
line 375 - kHz (not KHz)
line 391 - fig. 11 (fig. missing)
line 411 - 227x227 is not pixel size, but the size of the image in pixels
Subchapter 5.4 with equations 16 and 17 belongs to the methodological, not the experimental part of the article.
fig. 5 contains an example of convolution, which, however, is not explained in more detail, nor is it related to the process of signal processing from gearboxes.
fig. 14 presents prediction validation incomprehensible. Try using a different symbol (colour).
How is the pre-training process or processing of vibration signals from the gearbox related to revolutions, or by the number of gear teeth?
Why were the vibration sensors oriented to the vertical axis? Did you test this algorithm also when evaluating horizontal, or axial vibrations?

Reviewer 4 Report

1) It is not clear why the whale optimization algorithm has been used in the manuscript. The authors need to motivate the use of this algorithm. What are the features of the problem that make WOA an appropriate approach?

2) Literature review is too poor. In particular, there are multiple recent and relevant publications on wind turbine gearbox that could be used to enhance the literature review. For instance, consider discussing the following publications:

[R1]. "An Improved LightGBM Algorithm for Online Fault Detection of Wind Turbine Gearboxes", 2020. [https://doi.org/10.3390/en13040807]

[R2]. "Analysis and detection of a wind system failure in a micro-grid", 2016. [https://doi.org/10.1063/1.4960190]

[R3]. "Dual-Enhanced Sparse Decomposition for Wind Turbine Gearbox Fault Diagnosis", 2019. [http://doi.org/10.1109/TIM.2018.2851423]

3) Sections 2 and 3 are in total 7 pages. Since these sections summarize well-known concepts, they could be shorten.

4) It is not clear what kinds of faults are considered in the submitted manuscript. Also, how can one distinguish faults from each other?

5) Table 2: Sample size is too small. Also, samples are gathered only at one specific speed (800rpm). I am not sure if one can design a comprehensive fault detection scheme based upon this minimal observation. 

6) I failed to understand how the impact of noises is eliminated/treated. 

Round 2

Reviewer 3 Report

Mistakes are corrected, and questions are answered in the proper way. In my opinion, the article can be published.

Reviewer 4 Report

No further comment!